# Bile Salt Hydrolase-Competent Probiotics in the Management of IBD: Unlocking the “Bile Acid Code”

**DOI:** 10.3390/nu14153212

**Published:** 2022-08-05

**Authors:** Raffaella Maria Gadaleta, Marica Cariello, Lucilla Crudele, Antonio Moschetta

**Affiliations:** 1Department of Interdisciplinary Medicine, University of Bari “Aldo Moro”, Piazza Giulio Cesare 11, 70124 Bari, Italy; 2INBB National Instituto for Biostructure and Biosystems, Viale delle Medaglie d’Oro 305, 00136 Rome, Italy

**Keywords:** bile acids, BSH-competent bacteria, gut microbiota, intestinal inflammation, probiotics

## Abstract

Bile acid (BA) species and the gut microbiota (GM) contribute to intestinal mucosa homeostasis. BAs shape the GM and, conversely, intestinal bacteria with bile salt hydrolase (BSH) activity modulate the BA pool composition. The mutual interaction between BAs and intestinal microorganisms also influences mucosal barrier integrity, which is important for inflammatory bowel disease (IBD) pathogenesis, prevention and therapy. High levels of secondary BAs are detrimental for the intestinal barrier and increase the intestinal inflammatory response and dysbiosis. Additionally, a lack of BSH-active bacteria plays a role in intestinal inflammation and BA dysmetabolism. Thus, BSH-competent bacteria in probiotic formulations are being actively studied in IBD. At the same time, studies exploring the modulation of the master regulator of BA homeostasis, the Farnesoid X Receptor (FXR), in intestinal inflammation and how this impacts the GM are gaining significant momentum. Overall, the choice of probiotic supplementation should be a peculiar issue of personalized medicine, considering not only the disease but also the specific BA and metabolic signatures of a given patient.

## 1. Introduction

The entire gastrointestinal tract hosts the gut microbiota (GM), currently considered as an additional essential organ for living organisms. The GM’s physiological functions range from fermentation of non-digestible dietary fibers, to production of critical metabolites and immune-modulatory functions. The whole GM consists of different types of microorganisms, including bacteria, virus, fungi and even parasites. In the last two decades, studies have consistently shown the involvement of a dysbalanced gut bacterial composition in the pathogenesis of several diseases, while the role of virus and fungi is just starting to emerge. Intestinal bacteria could be categorized in six main phyla: Bacteroidota and Firmicutes make up to 90% of the total bacterial composition, while the remaining part is composed of Proteobacteria, Fusobacteria, Actinobacteria and Verrucomicrobia [1]. The host organism and its resident microbiota are in a constant mutual communication, regulated by sophisticated rules. A disproportionate number, reduced diversity or unbalanced communities of microorganisms (namely dysbiosis) decrease the ability of the host to resist infections, absorb nutrients, synthesize vitamins, and/or carry out peculiar metabolic activities [2], thereby contributing to the onset and progression of disease of the gut–liver axis [3], inflammatory [4], metabolic or autoimmune conditions [5] and even neurodegenerative disorders [6,7,8]. Despite the ‘big unknown’ in the GM field, it is universally accepted that the microbiome is unique for each individual and substantial differences are present even in identical twins [9]. In fact, the GM composition is not only determined genetically, but it is profoundly impacted by multiple factors including epigenetics, nutrition, drug administration, physical exercise, and general lifestyle.

In terms of metabolic functions, the GM produces short-chain fatty acids (SCFAs) [10] and synthesize vitamins, e.g., vitamin K and constituents of the vitamin B [11,12,13]. SCFAs, propionate, acetate and butyrate are produced by fiber fermentation and are involved in several metabolic functions, ranging from supplying energy and trophic factors to colonic cells [14], to modulating T regulatory cell functions [15,16] and exerting crucial physio-metabolic effects on several organs, including the brain [17,18]. Within the metabolic functions, a powerful emerging line of communication in organisms’ physiology is the relationship between the bile acid (BA) pool and the GM, as species-specific strains of bacteria are capable of metabolizing BAs [19].

## 2. Bile Acids: Between Digestion and Metabolic Signaling

BAs are digestive amphipathic compounds produced in the liver from the oxidation of cholesterol in two multistep biosynthetic pathways—the classic and the alternative pathway—leading to the production of primary BAs, mainly chenodeoxyxholic (CDCA) and cholic acid (CA) in humans, and CA and beta-muricholic acid (β-MCA) in mice. The classic pathway, regulated by the rate-limiting enzyme cholesterol 7 alpha-hydroxylase (Cyp7a1), produces CDCA and CA, while the alternative pathway is responsible for the production of intermediate oxidized cholesterol metabolites that are afterwards converted mainly to CDCA in hepatocytes. Before active secretion of BAs into the canalicular lumen, primary BAs are conjugated at the C24 carboxyl group with the aminoacids taurine or glycine [20]. This amidation process is very efficient and is important to make BAs less hydrophobic and, therefore, less cytotoxic, and more readily secretable into bile. Thus, the majority of BA species present in bile are conjugated. Once produced, BAs are stored in the gallbladder and, after a meal, they are secreted into the duodenum, where they flow and aid the absorption of lipophilic nutrients. After their journey along the intestinal tract performing their biological functions, BAs are then reabsorbed via combined mechanisms of active transport in the distal ileum and passive absorption along the whole intestine and are then re-circulated via the portal vein to the liver [21]. This process, known as enterohepatic circulation, occurs between 4 and 12 times per day in humans. Approximately 5% of BAs enter the colon, where they are mainly biotransformed by the GM and lost through feces. BA biotransformation carried out by the GM include subsequent steps of deconjugation, oxidation, epimerization, and also esterification and desulfatation [22,23]. Unconjugated BAs are generally more hydrophobic than the corresponding conjugated forms, and secondary BAs are more hydrophobic than the corresponding primaries; thus, unconjugated BAs and secondary BAs are more likely to cause noxious effects. A very recent work has shown that deletion of the gene encoding the enzyme catalyzing BA conjugation, the bile acid-CoA: amino acid N-acyltransferase (BAAT), causes metabolic changes, such as elevated plasma transaminases, lower lipid-soluble vitamin levels, higher fecal lipid content and lower intestinal BA levels, very similar to those reported in human BAAT mutations, together with an altered intestinal microbial composition [24].

The physiological effects of BAs have historically been attributed to their capacity of forming fat-solubilizing micelles aimed at aiding the absorption of dietary fats. Nowadays, it has become clear that in addition to this physico-chemical function, BAs behave as hormone-like signaling molecules and also interact with the GM. Therefore, the size and composition of the circulating BA pool play a crucial role not only in digestion, but also in regulating metabolic activities and shaping the gut microbial community [25]. Primary BAs exploit their endocrine function via binding to their cognate Farnesoid X Receptor (FXR) [26,27,28]. FXR is a nuclear receptor and transcription factor, highly expressed in the small intestine and liver acting as the master regulator of BA homeostasis. In fact, once activated by BAs, FXR regulates tissue-specific gene networks, orchestrating their synthesis, transport and metabolism (reviewed in [29]) in the gut–liver axis. In particular, via a negative feedback loop, activated FXR inhibits *de novo* BA hepatic synthesis, via the transactivation of one of its main intestinal targets, the enterokinefibroblast growth factor 15/19 (FGF15/19 in mouse and human, respectively) [30]. FGF15/19 is then secreted into the portal circulation and travels to the liver, where it binds to its cognate co-receptor heterodimer consisting of the fibroblast growth receptor 4 (FGFR4) and Klotho-beta (KLB), initiating a phosphorylation signaling cascade, ultimately resulting in CYP7A1 inhibition [31,32,33], hence suppressing *de novo* hepatic BA synthesis. In addition to this well-characterized pathway, BA-activated FXR also modulates BA cytotoxicity, aiming at creating a balanced BA pool that in physiological conditions and under proper nutritional cues does not harm the intestinal cell physiology. BA-activated FXR has also been shown to have anti-inflammatory [34,35,36] and antitumorigenic properties [37,38,39], and actively participates in the maintenance of intestinal barrier integrity [34,40]. Additionally, FGF19 was shown to have similar functions in mice and induce a reshape of the gut microbial community in the presence of an inflammatory trigger [41]. The ability of FXR and FGF19 to shape the circulating BA pool size and composition is critical for the relation BAs have with the GM community.

## 3. Bile Acid Deconjugationand BSH-Competent Microorganisms

The integrated metabolism of the BA pool and GM is among the most complex transgenomic biochemical interactions between the host and symbiotic microorganisms inhabiting it [42]. If one considers BA modulatory activities on cholesterol, triglycerides, glucose and energy homeostasis (reviewed in [43]), it is intuitive that gut bacteria-dependent BA biotransformation have crucial biological implications. Primary BAs synthesized and conjugated to glycine or taurine before secretion into the bile canaliculi are then transformed into secondary BAs, mainly deoxycholic acid (DCA), lithocolic acid (LCA) and β-muri-deoxycholic acid (β-MDCA; in mice) via bacteria with a peculiar activity: the bile salt hydrolase (BSH) activity [22,44,45]. The BSH catalyzes the “gateway” reaction in the bacterial metabolism of conjugated BAs hydrolyzing the C-24 N-acyl bond conjugating BA to glycine or taurine and is a prerequisite for the subsequent 7α/β-dehydroxylation [44,46,47,48] and 7α-dehydrogenation, ultimately resulting in BAs deconjugation and epimerization. GM-dependent BA metabolization mostly occurs in the terminal ileum and proximal colon, the focal colonization site of bacterial strains with BSH activity [22].

BSH activity was found in all major bacterial groups and archaeal species in the human gut [48]. Gram-positive gut bacteria, including strains of *Lactobacilli* [49,50], *Bifidobacteria* [51,52,53], *Clostridium* [54,55] and *Enterococcus* [56], display a highly heterogeneous distribution of BSH. On the contrary, BSH in Gram-negative bacterial strains has only been found in members of the phylum Bacteroidetes [48,57]. Recently published next-generation sequencing data on the human GM isolated from eleven populations from six continents unfolded the prevalence of BSH in 12 different phyla, including Bacteroidetes and Firmicutes [58]. Several strains, including *Clostridium scindens*, *Clostridium hiranonis*, and *Clostridium hylemonae*, have been discovered to be involved in the production of LCA and DCA in humans [23]. *Lactobacilli* [44,59], *Bifidobacterium longum* [53,60] and *Pediococcus* [61] have been shown to be BSH-competent species. Furthermore, pathogens such as *Clostridium perfringens* [62], *Listeria monocytogenes* [63], *Brucella abortus* [64], *Enterococcus faecalis* [65], and *Xanthomonas* [66] present with BSH activity. Last but not least, interesting findings have shown that bacteria isolated from marine sediments (*Methanosarcina acetovorans*) [48] and Antartican lakes (*Planococcus antarticus*) [67] display BSH activity.

In addition to the metabolic activities due to BSH-competent bacteria, there are other enzymatic biotransformations performed by gut microbes, such as the 7α/β-dehydroxylation, pyridine nucleotide-dependent hydroxysteroid dehydrogenization (operated by HSDHs enzymes), and conjugation of the cholate backbone with the aminoacids tyrosine, leucine and phenylalanine. The 7α/β-dehydroxylation is a multistep biotransformation carried out on free BAs and performed by a range of BA-inducible (bai) and bai-like genes [68]. Bai genes convert CA and CDCA to DCA and LCA, respectively, and have been identified in multiple species, mostly in *Faecali catena contorta S122*, *Clostridium hiranonis* and *Clostridium scinden* in humans [69] and *Eggerthella lenta* [70,71]. HSDHs performs the oxidation/reduction of hydroxy groups at the carbon atoms in positions 3, 7, and 12 of BAs [72], while the amide conjugation of cholate with aminoacids was recently found to be associated with *Clostridium bolteae* in mice fed a high-fat diet [73]; however, it is not currently clear what the function of such modified BA is.

FXR is differentially activated by the different BA species and their metabolites. Chenodeoxycholic Acid (CDCA) is the major FXR activator, followed by deoxycholic acid (DCA), lithocolic acid (LCA) and cholic acid (CA) [74]. Therefore, intestinal microorganisms not only act directly on BA species, but they also indirectly influence FXR transcriptional activity via the modulation of BA metabolism [75]. As a consequence, BA transformation by BSH-competent bacteria shapes the magnitude of BA-induced FXR activation in the gut [75,76,77,78]. Specifically, the major metabolic action of this specific group of bacteria is therefore to deconjugate intraluminal BAs, thus reducing the amount of conjugated BA (T/GCA and T/GCDCA) that can be absorbed by the enterocytes via the intestinal bile acid transporter (IBAT). These events translate into a de-activation of intestinal FXR (due to lower amounts of intracellular ligands) coupled with a significant increased fecal BA loss (up to 3 fold) (Figure 1). In this respect, one could speculate that the role of BSH-competent enteric bacterial would be similar to that of cholestyramine, which is able to bind intraluminal BA and increase their fecal release. Intriguingly, resins such ascholestyramine are nowadays used in clinic in BA-induced diarrhea in IBD and eventually BSH probiotics could complement this therapeutic strategy.

We have previously shown that enrichment of the gut milieu with BSH-competent species, such as *Lactobacilli* and *Bifidobacteria*, with VSL#3 probiotic administration in mice limits the availability of efficient FXR ligands and produces significant alterations in BA homeostasis that translates into enhanced fecal BA excretion, hepatic BA synthesis and biliary output [79], similar to what happens in mice receiving the BA sequestrant coleselavam [80]. Intriguingly, this phenomenon is abrogated in both Fxr- and Fgf15-null mice [79], indicating that the activation of the Fxr-Fgf15 duo is required for probiotic BSH-competent bacteria to promote fecal unconjugated BA loss and hepatic BA synthesis. For all these reasons, understanding how bacterial species affects BA metabolism and the FXR-FGF15/19 duo in the gut liver axis is a very relevant question gaining significant momentum from studies demonstrating that dysbiosis and BA dysmetabolism are associated with intestinal diseases.

## 4. Bile Acid-Dependent Shaping of Intestinal Microbial Community

The mutual interaction between the bileome and intestinal microorganisms has a direct influence on the integrity of the intestinal epithelial barrier. In fact, BAs flowing along the intestine control bacterial overgrowth and protect against intestinal mucosal damage. In particular, high levels of hydrophobic BAs have a direct antimicrobial activity mainly causing damage to the bacterial membrane [81,82]. On the one hand, Gram-negative bacteria, especially those belonging to Bacteroides, have been found to be more sensitive to free BAs [70,83]; on the other hand, Gram-positive bacteria seem to be more sensitive to tauro- and glyco-conjugated and primary BAs [59].

The tight interconnection between the GM and BA metabolism goes beyond BA intrinsic antimicrobial properties, as BAs indirectly shape the composition of GM by promoting the expression of genes fostering innate defense via FXR activation [40]. A lower level of BAs as a consequence of clinically and/or experimentally induced liver injuries can trigger intestinal bacterial overgrowth [84,85,86]. For example, surgical closure of the common bile duct in rodents results in bacterial overgrowth and translocation across the intestinal epithelial barrier, disruption of its integrity, and systematic infection that can be reversed by BA administration [87,88]. In the presence of an obstruction of bile flow, abnormal hepatic BA accumulation occurs and intestinal FXR signaling is abrogated [40]. In addition, devoiding the intestinal lumen of BA has a detrimental effect on the intestinal microbial community and causes mucosal injury [89]. Administration of BAs to both rodents and humans in conditions of biliary obstruction [88,90,91,92], as well as FXR activation [40] or FGF19 administration [41] in the presence of intestinal inflammatory triggers, counteracts intestinal mucosal damage and bacterial overgrowth and translocation. Rcent data from our group show that in a murine model of obstructed BA flow, FXR activation by its novel specific ligand TC-100 prevents intestinal mucosal damage, goblet cells and mucus depletion, and increases the expression of tight junctions by the mutual modulation of the GM and BA pool size and composition [93]. Both the peculiar enrichment of *Akkermansia muciniphila* and associated reduction in CA levels contribute to these beneficial effects [93]. BA-dependent activation of the Fxr-Fgf15/19 axis and the consequent inhibition of Cyp7a1, hence BA synthesis, is affected in rodents treated with antibiotics, germ-free and gnotobiotic animals [75,94,95,96,97,98] and mice lacking Fxr display ileal bacterial overgrowth [40] and a compromised epithelial barrier already at basal state [34,40].

Administration of the FXR-specific agonist obeticholic acid (OCA) in healthy volunteers was shown to inhibit endogenous BA synthesis while concomitantly enriching the GM with a number of Gram-positive strains, namely *Streptococcus thermophilus*, *Lactobacillus casei* and *paracasei*, *Bifidobacterium breve*, and *Lactococcus lactis*, boosting the notion that changes in the endogenous BA pool size and composition result in an FXR-dependent reshaping of the gut microbial composition [99]. In a mouse model of spontaneous colorectal cancer onset, the Apc^min/+^ mouse model, administration of a diet supplemented with CA increased the abundance of opportunistic pathogens such as *Prevotella* and *Desulfovibrio* while decreasing the presence of beneficial bacteria, including *Ruminococcus*, *Lactobacillus*, and *Roseburia* [100]. A similar diet administration in rats, induced alterations in the composition of the GM; the phylum of Firmicutes predominated at the expense of Bacteroidetes and an expansion of several strains in the classes *Clostridia* and *Erysipelotrichi* occurred [76]. Concomitantly, the administered CA was efficiently transformed into DCA by a bacterial 7α-dehydroxylation reaction [76], suggesting that DCA could boost the reshaping of the GM composition due to its powerful antimicrobial activity. In line with this, mice fed with a diet supplemented with DCA display a lower abundance of BSH-competent *Lactobacillus*, *Clostridium XI*, and *Clostridium XIV* and a concomitant enriched presence of *Parabacteroides* and *Bacteroides* compared to controls [101]. Taken together, these data demonstrate that BA supplementation reshapes the composition of gut microbiota.

## 5. Bile Salt Hydrolase Activity and Probiotics

Dietary supplementation with probiotics was demonstrated to modify not only hosts’ BA pool but also their metabolism. BSH-competent bacteria in probiotic formulations are being actively studied for their BA composition-modifying properties and their ability to reduce serum cholesterol levels [102]. In fact, BSH-dependent BA deconjugation decreases the recycling of BAs promoting their *de novo* synthesis, thus reducing serum cholesterol levels [102]. Additionally, bacteria with BSH activity metabolize aminoacids produced from deconjugation as carbon, nitrogen and energy sources [44]. Using germ-free and conventionally raised mouse models, Joyce et al. demonstrated that gastrointestinal expression of BSH promotes BA deconjugation with concomitant alteration of the host’s immune homeostasis, energy metabolism and circadian rhythm [103]. In conventionally raised mice, BSH activity reduced weight gain, serum cholesterol, and hepatic triglyceride levels [103]. Expression of cloned BSH enzymes in the gastrointestinal tract of gnotobiotic or conventionally raised mice exhibited a gene expression profile characterized by Clock and Arntl induction, genes involved in circadian rhythm regulation and energy expenditure [103]. Mukherji et al. observed that the GM, Toll-like receptor expression on epithelial cells and circadian rhythm are tightly connected and perturbations of this interaction promote the development of a pre-diabetic syndrome [104]. Administration of the BSH-competent *Lactobacillus reuteri NCIMB 30242*^®^ in human healthy volunteers improves unconjugated plasma BA profile highlighting the ability of this specific probiotic to modulate BA metabolism in humans [102]. Several studies conducted in mice and humans demonstrated the involvement of probiotics with *Lactobacillus* species such as *Lactobacillus fermentum K73* and *Lactobacillus plantarum 299v* in the reduction of body weight and cholesterol levels due to their BSH competence [102,105]. Interestingly, it was shown that the yeast strain *Saccharomyces Boulardii* has BSH activity in vitro [106]. Daily administration of this yeast in hamsters fed a high-cholesterol diet decreased total plasma cholesterol levels and modified the GM composition [107]. In humans, administration of *Saccaromices Boulardii* for 8 weeks reduced remnant lipoprotein concentration, a predictive biomarker and potential therapeutic target in the prevention/treatment of coronary artery disease [108]. Additionally, mice administered with the probiotic VSL#3, containing the BSH-competent *Lactobacillus acidophilus* and *Bifidobacterium infantis* strains, modulates the BA profile by upregulating hepatic *de novo* BA synthesis [79] and their fecal excretion [79].

## 6. BSH-Competent Bacteriaand Probiotics Administrationin Inflammatory Bowel Disease

Patients affected by inflammatory bowel diseases (IBD) display low levels of BSH activity in the gut microbiota [109,110]. IBD is a chronic inflammatory condition of the gastrointestinal tract characterized by a dysregulation of the gut mucosal immune functions and a dysbiotic GM occurring in genetically susceptible hosts. IBD encompasses two major phenotypes: ulcerative colitis (UC) and Crohn’s disease (CD), which present with overlapping symptoms, despite being two distinct pathologies.

Previously published data have shown how secondary BAs are detrimental and co-responsible for the disruption of intestinal barrier integrity, increase of the inflammatory response and dysbiosis [101,111,112]. Interestingly, rectal instillation of DCA/LCA in different murine models resulted in anti-inflammatory benefits, partially due to TGR5 signaling [113], indicating a dual pro- and anti-inflammatory effect of DCA in different experimental contexts and administration routes. This strongly suggests that caution should be taken when translating mouse data to IBD patients’ bedside.

Analysis of data collected on an IBD multiomics database (The Integrative Human Microbiome Project), revealed that BAs are a prominent component of the network in relation to changes in the microbiome [114]. In particular, primary BAs (specifically CA, TCA and GCA) levels were higher, while secondary BAs levels were lower [114]. This was associated with an enrichment of *Roseburia* and strongly indicated a lack of BSH-competent bacteria IBD-related dysbiosis [114]. Franzosa et al. also demonstrated that levels of primary BAs were higher in IBD patients and potentially associated with the coherent microbial groups, while secondary BA levels were lower in CD patients [115]. In line with this, patients with UC pouch displayed low levels of secondary BAs and high levels of CDCA and this was consistently associated with lower levels of *Ruminococcaceae* and bai genes [113], indicating once again a lack of BSH-competent bacteria, hence a dysbiosis-dependent disturbance of BA metabolization. Beside unravelling the intertwined relation between BA species and BSH-competent bacteria in IBD pathogenesis, its clinical and therapeutic relevance is also being actively studied. Omics data revealed that specific GM composition in IBD patients predicts early remission response to anti-cytokine and anti-integrin therapies, via gut bacterial clusterization associated with secondary BAs enrichment [71]. Probiotic supplementation in these patients may represent a useful tool for the prevention, treatment and remission of the acute phase. In IBD, probiotics have a beneficial impact when are able to modulate the immune response, gut barrier integrity and GM composition [116]. Several studies have been conducted in animal models and IBD patients to find suitable probiotics in IBD, but a universal consensus in not available yet.

Data on the GM composition of IBD patients indicated a lower level of Firmicutes and a higher level of Proteobacteria [117]. Consistent with this, recent data have shown that BSH-competent Proteobacteria are increased, while BSH-competent Firmicutes are decreased in IBD patients [118]. In vitro studies and mouse models have shown that *Faecalibacterium prausnitzii* and *Lactobacillus rhamnosus CNCM I-3690* ameliorate gut barrier integrity by increasing the expression of occludin and cadherin proteins [119,120]. Experimental colitis caused by Trinitrobenezenesulphonic acid (TNBS), increased *Escherichia coli* and *Clostridium* strains and reduced *Bifidobacterium* and *Lactobacillus* populations [121]. On the contrary, *Bifidobacterium* supplementation in murine experimental colitis, seems to have beneficial effects and decrease inflammation [122,123,124,125]. *Bifidobacterium infantis* administration reduced symptoms of colitis such as colonic edema and weight-loss, protected the epithelial cell layer, prevented goblet cell loss and modulated cytokine levels, specifically increasing the anti-inflammatory cytokine IL-10 and decreasing the pro-inflammatory one IL-1β [123,125,126]. In Sodium Dextran Sulphate (DSS)-induced murine colitis, administration of *Bifidobacterium animalis* and *Bifidobacterium longum* ameliorated inflammatory symptoms, preserved the colonic structure and decreased intestinal epithelial cell apoptosis and TNF-α levels [122,124]. Furthermore, in a TNBS-induced colitis mouse model, the influence of *Bifidobacterium animalis* and *Lactobacillus reuteri* administration on the immune system was studied [127]. Authors observed that *Bifidobacterium animalis* beneficial effects on the host were linked to the bone marrow-derived dendritic cell development and IL-17A secretion, whereas *Lactobacillus reuteri* administration promoted tolerogenic dendritic cells and Tregs population development [127]. Interestingly, two different formulations of VSL#3 have shown divergent metabolic and anti-inflammatory activities in mice subjected to DSS and TNBS colitis [128].

Clinical trials are ongoing in IBD patients with the aim to evaluate the effects of probiotics on the disease. In UC patients, several studies indicated that different *Bifidobacterium* strains are able to reduce pro-inflammatory interleukins and C-reactive protein (CRP) levels, sustaining the remission phase [129,130,131,132,133]. On the other hand, in CD patients, the administration of *Saccaromyces boulardii* promoted the maintenance of the remission phase [134]. Probio-Tec AB25, a cocktail of *Lactobacillus acidophilus* strain *LA-5* and *Bifidobacteriumanimalis subsp. lactis BB12*, was administered to 32 UC patients, with no beneficial effects in maintaining remission [135]. VSL#3 is among the most common probiotic mixtures of proven efficiency on human IBD, and it contains different *Lactobacilli*, *Bifidobacteria* and *Streptococcus thermophilus* strains [136,137]. In a double-blind placebo-controlled trial, patients with the active form of UC received VSL#3 twice daily for 12 weeks. Authors observed remission characterized by the reduction of stool frequency and rectal bleeding [136]. In another clinical trial, VSL#3 was used as an adjuvant added to patients’ pharmaceutical treatment for 8 weeks. A reduction in rectal bleeding was observed, although no differences were found in endoscopic inflammatory scores, highlighting the importance of long-term use of probiotics [137]. Furthermore, the administration of VSL#3 in children with UC determined a significant remission of patients due to a reduction of IFN-γ and TNF-α levels together with a modulation of the GM [138]. Conversely, VSL#3 treatment in CD patients does not seem to have clear benefits [139], although further research is warranted. In these patients, supplementation with *Faecalibacterium prausnitzii*, instead, exhibited anti-inflammatory effects [140]. Table 1 and Table 2 summarize the ongoing and/or completed clinical trials in IBD patients administered with either probiotics or microbial metabolites.

## 7. Conclusions and Discussion

BA species and intestinal microorganisms both contribute to intestinal fitness, are tightly interconnected and communicate directly with each other and indirectly via mainly FXR and related signaling pathways. Despite the evident association between BSH-competent bacteria and BA pool composition, metagenomic analysis of IBD patients’ GM revealed that not all BSH-competent species are created equal and the fact that any GM metabolic activity is strainspecific should always be remembered. Therefore, further studies on probiotics administration and their effect on gut microbial reshaping and consequent BA metabolism modulation are needed. At the same time, these studies should be paralleled by the generation of data on how microbial reshaping can be achieved dueto the modulation of the BA pool size and composition, as it occurs when FXR- or FGF19-based drugs are used. Clinical use of probiotics containing BSH-competent species is currently ongoing in patients with IBD, pouchitis and diarrhea and could be also relevant in patients with leaky gut secondary to liver disease (such as intrahepatic cholestasis) or colorectal cancer. On the other hand, a few considerations about limitations and potential adverse effects related to the indiscriminate use of probiotics resulting in high levels of unconjugated bile acids, lipid malabsorption and steatorrhea should be made. Elevated concentrations of DCA and LCA, produced from unconjugated BAs, are considered a risk factor for colorectal cancer [149]. Experimental evidence hasdemonstrated that BSH activity promotes gut colonization by pathogenic bacterial strains such as *Listeria monocytogenes* and *Brucella abortus* [63]. Moreover, mice and humans display different BA profiles characterized by high levels of tauro-conjugated BAs and high levels of glyco-conjugated BAs, respectively [59,74]. In this scenario, previous data on the species-specific difference affecting the translation of data from mice to humans and confounding the selection of probiotics are actively being validated in clinical trials. The field is new and a greatereffort should be made in integrating the enormous amounts of data generated by all the available omic platforms. Overall, the choice of probiotic supplementation should be considered a peculiar issue of the personalized medicine, considering not only the disease but also the specific BA and metabolic signature of a given patient.

## Figures and Tables

**Figure 1 nutrients-14-03212-f001:**
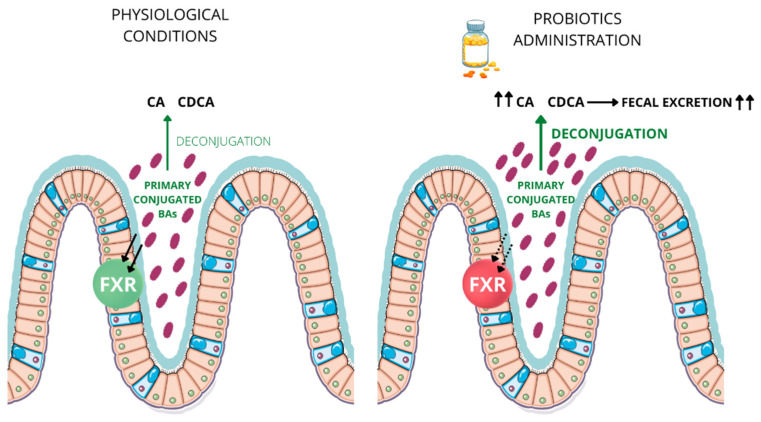
BSH-competent bacteria activity in probiotics. In physiological conditions, there is a balance between the input of conjugated BA taken up in the enterocytes and activating FXR and the unconjugated BA fecal output. Enriching the intestinal milieu with BSH-competent intestinal microorganisms induces BA transformation that, in turn, shapes the magnitude of FXR activation in the enterocytes. This translates into increased fecal BA excretion and lower FXR transcriptional activation. (Bas = bile acids; CA = cholic acid; CDCA = chenodeoxycholic acid; FXR = Farnesoid X Receptors; green FXR indicates activated FXR; red FXR indicated de-activated FXR; purple globus indicates BSH-competent bacteria; black arrows pointing up indicate an increase of CA, CDCA and fecal excretion, bold green arrow pointing up—right panel—indicates increased deconjugation).

**Table 1 nutrients-14-03212-t001:** Clinical Trials exploiting the use of Probiotics in IBD.

Trial Identifier	Trail Phase (Status)	Title	Conditions	Interventions
NCT03266484	Active, not recruiting	Effect of a Probiotic Mixture on the Gut Microbiome and Fatigue in Patients With Quiescent Inflammatory Bowel Disease	Crohn’s Disease Ulcerative Colitis	- Dietary Supplement: Probiotic Mixture - Dietary Supplement: Placebo
NCT01765998	Unknown	The Effect of Probiotics on Exacerbation of Inflammatory Bowel Disease Exacerbation (Crohn’s Disease)	Crohn’s Disease	- Drug: Probiotic - Drug: Placebo
NCT01765439	Active, not recruiting	The Effect of VSL#3 Probiotic Preparation on the Bile Acid Metabolism in Patients With Inflammatory Bowel Disease	Crohn’s Disease Ulcerative Colitis	Dietary Supplement: VSL#3
NCT00175292	Completed	A Randomized Controlled Trial of VSL#3 for the Prevention of Endoscopic Recurrence Following Surgery for Crohn’s Disease	Crohn’s Disease Inflammatory Bowel Disease	Drug: Probiotic—VSL#3
NCT01078935	Unknown	The Effect of Probiotics on the Rate of Recovery of Inflammatory Bowel Disease Exacerbation, Endothelial Function, and Markers of Inflammation	Crohn’s Disease Ulcerative Colitis	- Dietary Supplement: probiotics - Dietary Supplement: placebo
NCT01173588	Completed	Effect of Yogurt Added With Bifidobacteria and Soluble Fiber on Bowel Function	Inflammatory Bowel Disease	Other: yogurt added with bifidobacteria and soluble fiber (YBF)
NCT01479660	Unknown	Role of Healthy Bacteria in Ulcerative Colitis	Ulcerative Colitis	- Other: Control - Drug: Probiotic
NCT00510978	Unknown	Probiotics in GastroIntestinal Disorders (PROGID)	Crohn’s Disease Ulcerative Colitis	- Biological: Bifidobacterium infantis 35624 - Biological: Lactobacillus salivarius UCC118 - Biological: Placebo
NCT01632462	Unknown	A Prospective, Placebo Controlled, Double-Blind, Cross-over Study on the Effects of a Probiotic Preparation (VSL#3) on Metabolic Profile, Intestinal Permeability, Microbiota, Cytokines and Chemokines Expression and Other Inflammatory Markers in Pediatric Patients With Crohn’s Disease	Crohn’s Disease	Drug: VSL#3
NCT01548014	Unknown	The Effect of a Probiotic Preparation (VSL#3) Plus Infliximab in Children With Crohn’s Disease	Crohn’s Disease	- Dietary Supplement: VSL#3
NCT00374374	Completed	Treatment With Lactobacillus Rhamnosus and Lactobacillus Acidophilus for Patients With Active Colonic Crohn’s Disease	Crohn’s Disease	- Behavioral: Administration of probiotic
NCT03565939	Completed	Probiotic Treatment of Ulcerative Colitis With Trichuris Suis Ova (TSO) (PROCTO)	Ulcerative Colitis Chronic Moderate	- Biological: Trichuris suis ova - Biological: Placebo
NCT00114465	Completed	VSL#3 Versus Placebo in Maintenance of Remission in Crohn’s Disease	Crohn’s Disease	- Drug: VSL#3- Other: Placebo
NCT00944736	Completed	Effect of VSL#3 on Intestinal Permeability in Pediatric Crohn’s Disease	Crohn’s Disease	- Dietary Supplement: VSL#3 - Dietary Supplement: Placebo
NCT04842149	Recruiting	The Effects of Bifidobacterium Breve Bif195 for Small Intestinal Crohn’s Disease	Crohn’s Disease	- Dietary Supplement: Bif195 capsules - Dietary Supplement: Placebo capsules
NCT04223479	Completed	Effect of Probiotic Supplementation on the Immune System in Patients With Ulcerative Colitis in Amman, Jordan	Ulcerative Colitis	- Drug: Probiotic Formula Capsule - Drug: Placebos
NCT01698970	Completed	Effect of the Consumption of a Probiotic Strain on the Prevention of Post-operative Recurrence in Crohn’s Disease	Crohn’s Disease	- Other: 1-Freeze-dried Probiotics provided in capsule (150 mg) containing 1.0 × 10^10^ colony forming unit per capsule (test) - Other: 2-excipients (150 mg) in capsule (control)
NCT01772615	Completed	Treatment of Ulcerative Colitis With Ciprofloxacin and *E. Coli* Nissle	Ulcerative Colitis	- Drug: Ciprofloxacin - Dietary Supplement: *E. coli* Nissle
NCT02488954	Terminated	Interest of Propionibacterium Freudenreichii for the Treatment of Mild to Moderate Ulcerative Colitis (EMMENTAL)	Ulcerative Colitis	Other: Probiotics in the form of cheese portion
NCT04305535	Unknown	Impact of an Oligomeric Diet in Intestinal Absorption and Inflammatory Markers in Patients With Crohn Disease	Crohn Disease Absorption; Disorder: Protein Absorption; Disorder Absorption; Disorder: Fat Absorption; Disorder: Carbohydrate Malnutrition	- Dietary Supplement: Peptidic + Probiotic - Dietary Supplement: Peptidic + Placebo - Dietary Supplement: Polymeric + Placebo
NCT00305409	Completed	Synbiotic Treatment in Crohn’s Disease Patients	Crohn’s Disease	Drug: Synbiotic (Synergy I/B. longum)
NCT04804046	Recruiting	Synbiotics and Post-op Crohn’s Disease	Crohn’s Disease	- Dietary Supplement: Synbiotic - Other: Digestible Maltodextrin
NCT00803829	Completed	Synbiotic Treatment of Ulcerative Colitis Patients	Ulcerative Colitis	- Other: Synbiotic (Synergy1/B. longum)
NCT00951548	Completed	Food Supplementation With VSL#3 as a Support to Standard Pharmaceutical Therapy in Ulcerative Colitis	Ulcerative Colitis	- Dietary Supplement: VSL#3 - Dietary Supplement: Placebo
NCT00374725	Completed	Treatment of Ulcerative Colitis With a Combination of Lactobacillus Rhamnosus and Lactobacillus Acidophilus.	Ulcerative Colitis	Behavioral: Administration of probiotic (L. rhamnosus and L. acidophilus)
NCT03415711	Terminated	PRObiotic VSL#3^®^ for Maintenance of Clinical and Endoscopic REMission in Ulcerative Colitis	Ulcerative Colitis	- Dietary Supplement: VSL#3^®^ - Drug: MesalamineDrug: Placebo
NCT04102852	Recruiting	Lactobacillus Rhamnosus GG (ATCC 53103) in Mild-moderately Active UC Patients	Ulcerative Colitis Chronic Mild Ulcerative Colitis Chronic Moderate	Dietary Supplement: Lactobacillus rhamnosus GG ATCC 53103
NCT00367705	Unknown	VSL#3 Treatment in Children With Crohn’s Disease	Crohn’s Disease	- Dietary Supplement: VSL#3^®^ - Dietary Supplement: Placebo
NCT04969679	Completed	Additive Effect of Probiotics (Mutaflor^®^) in Patients With Ulcerative Colitis on 5-ASA Treatment	Ulcerative Colitis	- Drug: *E. coli* Nissle 1917 (Mutaflor^®^) - Drug: Placebo
NCT00268164	Terminated	Lactobacillus Acidophilus and Bifidobacterium Animalis Subsp. Lactis, Maintenance Treatment in Ulcerative Colitis	Ulcerative Colitis	Drug: lactobacilus acidophilus & bifidobacterium animalis/lactis

**Table 2 nutrients-14-03212-t002:** Clinical Trials exploiting the use of Microbial Metabolites in IBD.

Trial Identifier	Trail Phase (Status)	Title	Conditions	Interventions
**Short Chain Fatty Acids**				
NCT05456763	Completed	Butyrate in Pediatric Inflammatory Bowel Disease	IBD	- Dietary Supplement: sodium butyrate - Other: placebo
**Tryptophan Metabolites**				
NCT04089501	Completed	The Role of the Pregnane X Receptor (PXR) in Indole Signaling and Intestinal Permeability in Inflammatory Bowel Disease	IBD	- Diagnostic Test: Stool collection - Diagnostic Test: Biopsy collection
**Bile Acid Metabolites**				
NCT03724175	Recruiting	The Role of Secondary Bile Acids in Intestinal Inflammation	Ulcerative Colitis Pouchitis	- Drug: ursodiol (ursodeoxycholic acid, UDCA)
**Vitamines**				
NCT03145896	Unknown	The Correlation Between Anemia of Chronic Diseases, Hepcidin and Vitamin D in IBD Patients	IBD	Dietary Supplement: vitamin D
NCT03162432	Completed	High Dose Interval Vitamin D Supplementation in Patients With IBD Receiving Remicade	IBD Ulcerative Colitis Crohn Disease	Drug: Vitamin D3
NCT02076750	Completed	Weekly Vitamin D in Pediatric IBD	IBD Skin Pigmentation	Dietary Supplement: Vitamin D3 (cholecalciferol)
NCT00621257	Terminated Has Results [141,142]	Vitamin D Levels in Children With IBD	IBD Crohn’s Disease Ulcerative Colitis	- Dietary Supplement: ergocalciferol - Dietary Supplement: Cholecalciferol
NCT02256605	Completed	Vitamin D3 Supplementation in Pediatric IBD: Weely vs Daily Dosing Regimens	IBD	Dietary Supplement: Vitamin D-3
NCT04225819	Recruiting	Adjunctive Treatment With Vitamin D3 in Patients With Active IBD	IBD Crohn Disease Ulcerative Colitis Vitamin D3 Deficiency	- Dietary Supplement: Vitamin D3 - Other: Placebo
NCT03496246	Unknown	Vitamin D Status in Inflammatory Bowel Disease (vdsinibd)	Vitamin D Deficiency IBD	- Diagnostic Test: serum total 25(OH) vitamin D - Diagnostic Test: complete blood count (CBC) - Diagnostic Test: serum calcium level - Diagnostic Test: erythrocyte sedimentation rate (ESR) - Diagnostic Test: C-reactive protein (CRP) - Diagnostic Test: serum creatinine - Diagnostic Test: serum albumin level - Diagnostic Test: seum alanine aminotransferase - Diagnostic Test: serum potassium level - Diagnostic Test: serum phosphurus level
NCT04991324	Not yet recruiting	Cholecalciferol Comedication in IBD—the 5C-study (5C)	IBD	Drug: Vitamin D3
NCT04828031	Recruiting	Vitamin D Regulation of Gut Specific B Cells and Antibodies Targeting Gut Bacteria in Inflammatory Bowel Disease	IBD Ulcerative Colitis Crohn Disease	Drug: Vitamin D
NCT04331639	Recruiting	High Dose Interval Vitamin D Supplementation in Patients With Inflammatory Bowel Disease Receiving Biologic Therapy	IBD Crohn Disease Ulcerative Colitis Vitamin D Deficiency	Dietary Supplement: vitamin D3
NCT01877577	Completed	Supplementation of Vitamin D3 in Patients With Inflammatory Bowel Diseases and Hypovitaminosis D	Crohn’s Disease Ulcerative Colitis	Dietary Supplement: Vitamin D3
NCT00742781	Completed Has Results	Vitamin D Supplementation in Crohn’s Patients	IBD	Dietary Supplement: Vitamin D
NCT01121796	Unknown	Influence of Vitamin D on Inflammatory Bowel Disease Remission	IBD	- Dietary Supplement: Vitamin D - Other: Water or milk - Dietary Supplement: Vitamin D enriched milk
NCT01792388	Completed	Vitd and Barrier Function in IBD	Crohn’s Disease	- Dietary Supplement: Vitamin D - Dietary Supplement: Soya Bean oil
NCT00152841	Terminated	Effect of Iron and Vitamin E Supplementation on Disease Activity in Patients With Either Crohn’s Disease or Ulcerative Colitis	Crohn’s Disease Ulcerative Colitis Mild or Moderate Anaemia	- Drug: Iron supplement 300–600 mg/day - Drug: Vitamin E 800 IU
NCT00114803	Completed	Nasal Calcitonin in the Treatment of Bone Mineral Loss in Children and Adolescents With Inflammatory Bowel Disease	Ulcerative Colitis Crohn’s Disease	Drug: Calcitonin nasal spray (salmon)
NCT04913467	Recruiting	Effect of Ileocolonic Delivered Vitamins and an Anti-Inflammatory Diet on Crohn’s Disease and Healthy Volunteers	Crohn Disease	- Other: Groningen Anti-Inflammatory Diet (GrAID) - Dietary Supplement: ColoVit capsule - Other: ColoPulse-placebo capsule
NCT03718182	Unknown	Can Vitamin D Supplementation in People With Crohn’s Disease Improve Symptoms as an Adjunct Therapy?	Crohn Disease Vitamin D Deficiency	Dietary Supplement: Cholecalciferol
NCT02615288	Completed	High Dose Vitamin D3 in Crohn’s Disease	Crohn Disease	Dietary Supplement: Vitamin D3
NCT01692808	Completed	Bioavailability of Vitamin D in Children and Adolescents With Crohn’s Disease	Crohn’s Disease	- Drug: Vitamin D3 3000 UI daily - Drug: Vitamin D3 4000 UI daily
NCT02186275	Completed	The Vitamin D in Pediatric Crohn’s Disease	Crohn’s Disease	- Drug: Vitamin D3: 3000 or 4000 UI/day then 2000 UI/day - Drug: Vitamin D3 800 UI/day then 800 UI/day
NCT03999580	Recruiting	The Vitamin D in Pediatric Crohn’s Disease (ViDiPeC-2)	Crohn Disease	Drug: vitamin D3
NCT01235325	Completed	The Effect of Vitamin K Supplementation on Bone Health in Adult Crohn’s Disease Patients	Supplementation Bone Health Crohn’s Disease	- Dietary Supplement: phylloquinone (vitamin K1) - Dietary Supplement: placebo
NCT00132184	Unknown	Vitamin D Treatment for Crohn’s Disease	Crohn Disease	Drug: Vitamin D
NCT01369667	Completed	Vitamin D Supplementation in Adult Crohn’s Disease	Crohn Disease	- Dietary Supplement: Vitamin D3 - Other: Placebo
NCT01864616	Terminated	The Impact of Vitamin D on Disease Activity in Crohn’s Disease	Crohn Disease	Dietary Supplement: Vitamin D3
NCT00427804	Completed Has Results [143]	Tumor Necrosis Factor Decreases Vitamin D Dependant Calcium Absorption	Rheumatoid Arthritis Crohn’s Disease	Drug: calcitriol
NCT04308850	Not yet recruiting	Exploring the Effects of Vitamin D Supplementation on the Chronic Course of Patients With Crohn’s Disease With Vitamin D Deficiency	Crohn’s Disease Vitamin D Deficiency Vitamin D Supplement	Drug: Vitamin D drops
NCT04134065	Unknown	The Effect of Vitamin D in Crohn’s Disease	Crohn’s Disease Vitamin D Deficiency	- Drug: Vitamin D - Drug: Placebo oral capsule
NCT02704624	Unknown	Effects of Supplementation of Vitamin D in Patients With Crohn’s Disease	Crohn Disease Vitamin D Deficiency Fatigue Sarcopenia Muscle Weakness Disorder of Bone Density and Structure, Unspecified	- Dietary Supplement: Vitamin D - Other: Placebo
NCT02208310	Terminated Has Results [144,145,146,147]	Trial of High Dose Vitamin D in Patient’s With Crohn’s Disease	Crohn’s Disease Vitamin D Deficiency	- Drug: Cholecalciferol 10,000 IU - Drug: Cholecalciferol 400 IU
NCT03615378	Terminated	Maintenance Dosing of Vitamin D in Crohn’s Disease	Crohns Disease Vitamin D Deficiency	- Dietary Supplement: 5000 IU D3 - Dietary Supplement: 1000 IU D3 - Dietary Supplement: Placebo
NCT04309058	Unknown	Observation of the Effect of Vitamin D Supplementation on Chronic Course of Patients With Ulcerative Colitis Based on Vitamin D Receptor Fok I Gene Polymorphism	Ulcerative Colitis Vitamin D Deficiency Vitamin D Supplement	Drug: Vitamin D drops
NCT01046773	Terminated	Vitamin D Supplementation as Non-toxic Immunomodulation in Children With Crohn’s Disease	Crohn’s Disease Vitamin D Deficiency	Drug: Cholecalciferol
NCT04276649	Completed	A Retrospective Analysis: the Influence of Caltrate Supplement on the Effect of Mesalazine in Ulcerative Colitis	Ulcerative Colitis Vitamin D Deficiency Vitamin D Supplement	Drug: Caltrate
NCT04259060	Not yet recruiting	Hydroxocobalamin Approach for Reducing of Calprotectin With Butyrate for Ulcerative Colitis Remission	Ulcerative Colitis	- Drug: Hydroxocobalamin with Butyrate - Drug: Placebo with Butyrate
**Sulfur-Containing Metabolites**				
NCT01282905	Completed	Hydrogen Sulfide Detoxification and Butyrate Metabolism in Ulcerative Colitis	Ulcerative Colitis	/
NCT04474561	Completed	Reduced Sulfur Diet in Ulcerative Colitis Patients	Ulcerative Colitis Diet Habit	Other: Reduced sulfur diet
**Polyphenol Metabolites**				
NCT00718094	Completed Has Results [148]	Pilot Study of Green Tea Extract (Polyphenon E^®^)in Ulcerative Colitis	Mild to Moderately Active Ulcerative Colitis	- Drug: Polyphenon E^®^ - Drug: Placebo Oral Tablet

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
