# Peer review of "Bile Salt Hydrolase-Competent Probiotics in the Management of IBD: Unlocking the “Bile Acid Code”"

_nutrients, 2022, doi:10.3390/nu14153212_

Round 1

Reviewer 1 Report

1). This is an interesting review article on what is known about bile-salt hydrolase in bacteria of the GI track.

2. I do not think the "Bile Acids Code" in the title is appropriate and should be removed.

3. There is a very recent publication (Bandar Alrehali ... Yoon-Kwang Lee. Bile acid conjugation deficiency causes........Hepatology Communications) where scientists made KO mice that cannot conjugate BAs. That work is extremely important in informing the importance of conjugation is in BA functions. Hepatology Communications This really should be included.

4. There is also a recent review article on BAs by Choudhuri in Drug Metabolism Disposition 50:425, 2022. There is a lot of information that is relevant for the first few pages of your review, that will provide information to correct some errors in the present review.

5. Line 67: This is not true, it does not produce CDCA and CA in equal amount.

6. Line 68: Cyp1b does not regulate the alternative pathway

7, Line 128: 7a or 7alpha?

Author Response

1) This is an interesting review article on what is known about bile-salt hydrolase in bacteria of the GI track.

We thank the Reviewer for this comment.

2) I do not think the "Bile Acids Code" in the title is appropriate and should be removed.

We thank the reviewer for this remark. However, we think that the word bile acid should be somewhere included given the fact that BSH action involves directly bile acid metabolism. Therefore, as also discussed with the editors, we slightly modified the title into keeping the word “bile acids” thus we respectfully hope that the Reviewer would agree with this option.  

3) There is a very recent publication (Bandar Alrehali ... Yoon-Kwang Lee. Bile acid conjugation deficiency causes........Hepatology Communications) where scientists made KO mice that cannot conjugate BAs. That work is extremely important in informing the importance of conjugation is in BA functions. Hepatology Communications This really should be included.

We thank the reviewer for this suggestion. We have now included this reference in Paragraph 2.

4) There is also a recent review article on BAs by Choudhuri in Drug Metabolism Disposition 50:425, 2022. There is a lot of information that is relevant for the first few pages of your review that will provide information to correct some errors in the present review.

We thank the reviewer for his remark and apologise for the mistakes. We have read the suggested review and corrected the mentioned mistakes (see point 5 and 6).

5) Line 67: This is not true, it does not produce CDCA and CA in equal amount.

We thank the reviewer for this comment. We apologize for the wrong sentence and have now modified it.

6) Line 68: Cyp1b does not regulate the alternative pathway.

We thank the reviewer for this comment. We have now modified the sentence.

7) Line 128: 7a or 7alpha?

The enzyme is Cholesterol 7 alpha-hydroxylase and the acronym is Cyp7a1.

Reviewer 2 Report

The review hihglighted importance bile acid (BA) species and the gut microbiota (GM)/their products and their contribution to the intestinal homeostasis. Overall, this manuscript is well written and provide with a logical flow of  factors that in involved in IBD disease progression. I have some minor comments.

1-    Figures 1 should be improved. Some of the font size are very small and not readable and in consistent. Also, figure legend might be placed under the figure with abbreviations that used in figure.

2-    In addition to figure, maybe there could be a table summarizing the factors (BAs , microorganisms that produce the important factors, probiotics that used for IBD therapeutics) involved in intestinal homeostasis that affects disease progression with their activities e.t.c. in addition, maybe there could be some additional information the impact of the probiotics, microbial products, dysbiosis and their role on especially gut-liver or gut-brain axis.

3-    To me, there are too many references. Some of references that are very old might be eliminated.

4-    In page 5, lines 227-229, if published, the reference should be updated.

5-    Gene/protein/species names should be written according to nomenclature i.e. all species of microorganisms are not written according to nomenclature.

6-    it is  seemed that section 3 is divided into subsections but there is 3.2. but not 3.1. Numbering of sections should be clarified.

Author Response

The review highlighted importance bile acid (BA) species and the gut microbiota (GM)/their products and their contribution to the intestinal homeostasis. Overall, this manuscript is well written and provide with a logical flow of  factors that in involved in IBD disease progression. I have some minor comments.

1-    Figures 1 should be improved. Some of the font size are very small and not readable and in consistent. Also, figure legend might be placed under the figure with abbreviations that used in figure.

We thank the reviewer for this remark. We have improved Figure 1, as requested.

2-    In addition to figure, maybe there could be a table summarizing the factors (BAs, microorganisms that produce the important factors, probiotics that used for IBD therapeutics) involved in intestinal homeostasis that affects disease progression with their activities e.t.c. in addition, maybe there could be some additional information the impact of the probiotics, microbial products, dysbiosis and their role on especially gut-liver or gut-brain axis.

We thank the reviewer for raising this point. Following the Reviewer suggestion, we have now added two tables about probiotics and microbial metabolites used in clinical trials for IBD patients. Indeed, the manuscript sounds more complete. On a different angle, we respectfully did not add more information on the gut-brain axis as there is no more word space and it would be out of scope for this review that focuses on IBD.

3-    To me, there are too many references. Some of references that are very old might be eliminated.

We thank the Reviewer for this remark. We have now eliminated a few references, although this might not be immediately appreciable as we have added a couple of new references as requested by another reviewer. Moreover, we have added two new tables and updated one reference that was still in a preliminary format, as requested (see point 2 and 4). One of the table required the addition of 8 more references.

4-    In page 5, lines 227-229, if published, the reference should be updated.

We thank the Reviewer for pointing this out. The manuscript is now available online and we have updated the reference.

5-    Gene/protein/species names should be written according to nomenclature i.e. all species of microorganisms are not written according to nomenclature.

We thank the Reviewer for pointing this out. We have now corrected this.

6-    it is  seemed that section 3 is divided into subsections but there is 3.2. but not 3.1. Numbering of sections should be clarified.

We thank the reviewer for pointing this out. We have now corrected the numbering of the sections.